# Practical Two-Step Look-Ahead Bayesian Optimization

**Jian Wu**
wujian046@gmail.com

**Peter I. Frazier**[*]
School of Operations Research and Information Engineering
Cornell University
Ithaca, NY 14850
pf98@cornell.edu

## Abstract

Expected improvement and other acquisition functions widely used in Bayesian optimization use a "one-step" assumption: they value objective function evaluations assuming no future evaluations will be performed. Because we usually evaluate over multiple steps, this assumption may leave substantial room for improvement. Existing theory gives acquisition functions looking multiple steps in the future but calculating them requires solving a high-dimensional continuous-state continuous-action Markov decision process (MDP). Fast exact solutions of this MDP remain out of reach of today's methods. As a result, previous two- and multi-step lookahead Bayesian optimization algorithms are either too expensive to implement in most practical settings or resort to heuristics that may fail to fully realize the promise of two-step lookahead. This paper proposes a computationally efficient algorithm that provides an accurate solution to the two-step lookahead Bayesian optimization problem in seconds to at most several minutes of computation per batch of evaluations. The resulting acquisition function provides increased query efficiency and robustness compared with previous two- and multi-step lookahead methods in both single-threaded and batch experiments. This unlocks the value of two-step lookahead in practice. We demonstrate the value of our algorithm with extensive experiments on synthetic test functions and real-world problems.

## 1 Introduction

We consider minimization of a continuous black-box function $f$ over a hyperrectangle $\mathcal{A} \subseteq \mathbb{R}^d$. We suppose evaluations $f(x)$ are time-consuming to obtain, do not provide first- or second-order derivative information and are noise-free. Such problems arise when tuning hyperparameters of complex machine learning models [Snoek et al., 2012] and optimizing engineering systems using physics-based simulators [Forrester et al., 2008].

We consider this problem within a Bayesian optimization (BayesOpt) framework [Brochu et al., 2010, Frazier, 2018]. BayesOpt methods contain two components: (1) a statistical model over $f$, typically a Gaussian process [Rasmussen and Williams, 2006]; and (2) an acquisition function computed from the statistical model that quantifies the value of evaluating $f$. After a first stage of evaluations of $f$, often at points chosen uniformly at random from $A$, we behave iteratively: we fit the statistical model to all available data; then optimize the resulting acquisition function (which can be evaluated quickly and often provides derivative information) to find the best point(s) at which to evaluate $f$; perform these evaluations; and repeat until our evaluation budget is exhausted.

---

[*]Peter Frazier is also a Staff Data Scientist at Uber

The most widely-used acquisition functions use a one-step lookahead approach. They consider the *direct* effect of the evaluation on an immediate measure of solution quality, and do not consider evaluations that will be performed later. This includes expected improvement (EI) [Jones et al., 1998], probability of improvement (PI) Kushner [1964], entropy search (ES) [Hernández-Lobato et al., 2014, Wang and Jegelka, 2017], and the knowledge gradient (KG) [Wu and Frazier, 2016]. By myopically maximizing the immediate improvement in solution quality, they may sacrifice even greater gains in solution quality obtainable through coordinated action across multiple evaluations.

Researchers have sought to address this shortcoming through non-myopic acquisition functions. The decision of where to sample next in BayesOpt can be formulated as a partially observable Markov decision process (POMDP) [Ginsbourger and Riche, 2010]. The solution to this POMDP is given by the Bellman recursion [Lam et al., 2016] and provides a non-myopic acquisition function that provides the best possible average-case performance under the prior. However, the "curse of dimensionality" Powell [2007] prevents solving this POMDP for even small-scale problems.

The past literature [Lam et al., 2016, Osborne et al., 2009, Ginsbourger and Riche, 2010, González et al., 2016] instead approximates the solution to this POMDP to create non-myopic acquisition functions. Two-step lookahead is particularly attractive [Osborne et al., 2009, Ginsbourger and Riche, 2010, González et al., 2016] because it is substantially easier to compute than looking ahead more than two steps, but still promises a performance improvement over the one-step acquisition functions used in practice. Indeed, Ginsbourger and Riche [2010] argue that using two-step lookahead encourages a particularly beneficial form of exploration: evaluating a high uncertainty region benefits future evaluations; if the evaluation reveals the region was better than expected, then future evaluations evaluate nearby to find improvements in solution quality. This benefit occurs even if the first evaluation does not generate a direct improvement in solution quality. In numerical experiments, Osborne et al. [2009], Ginsbourger and Riche [2010] show that two-step lookahead improves over one-step lookahead in a range of practical problems.

At the same time, optimizing two-step acquisition functions is computationally challenging. Unlike common one-step acquisition functions like expected improvement, they cannot be computed in closed form and instead require a time-consuming simulation with nested optimization. Simulation creates noise and prevents straightforwand differentiation, which hampers optimizing these two-step acquisition functions precisely. Existing approaches [Osborne et al., 2009, Ginsbourger and Riche, 2010, González et al., 2016] use derivative-free optimizers, which can require a large number of iterations to optimize precisely, particularly as the dimension $d$ of the feasible space grows. (Numerical experiments in Osborne et al. [2009], Ginsbourger and Riche [2010] are restricted to problems with $d \leq 3$.) As a result, existing two- and multi-step methods require a prohibitive amount of computation (e.g., Lam [2018] reports that the method in Lam et al. [2016] requires between 10 minutes and 1 hour per evaluation even on low-dimensional problems). If sufficient computation is not performed, then errors in the acquisition-function optimization overwhelm the benefits provided by two-step lookahead and query efficiency degrades compared to a one-step acquisition function supporting precise optimization. Similar challenges arise for the multi-step method proposed in Lam et al. [2016]. This computational challenge has largely prevented the widespread adoption of non-myopic acquisition functions in practice.

**Contributions.** This article makes two key innovations unlocking the power of two-step lookahead in practice. First, we provide an estimator based on the envelope theorem for the gradient of the two-step lookahead acquisition function. Second, we show how Monte Carlo variance reduction methods can further reduce the computational cost of estimating both the two-step lookahead acquisition function and its gradient. These techniques can be used within multistart stochastic gradient ascent to efficiently generate multiple approximate stationary points of the acquisition function, from which we can select the best to provide an efficient approximate optimum. Together, these innovations support optimizing the acquisition function accurately with computation requiring between a few seconds and several minutes on a single core. Moreover, this computation can be easily parallelized across cores. It also scales better in the batch size and dimension of the black-box function compared with the common practice of using a derivative-free optimizer. An implementation is available in the Cornell-MOE codebase, `https://github.com/wujian16/Cornell-MOE`, and the code to replicate our experiments is available at `https://github.com/wujian16/TwoStep-BayesOpt`.

Our approach leverages computational techniques developed in the literature. The first is infinitesimal perturbation analysis [Heidelberger et al., 1988] and the envelope theorem [Milgrom and Segal, 2002],

previously used in Bayesian optimization to optimize the knowledge gradient aquisition function (which is myopic, as noted above) by Wu et al. [2017]. This built on earlier work using infinitesimal perturbation analysis without the envelope theorem to optimize the expected improvement acquisition function (also myopic) in the batch setting [Wang et al., 2016]. The second is a pair of variance reducton methods: Gauss-Hermite quadrature Liu and Pierce [1994] and importance sampling [Asmussen and Glynn, 2007]. Our paper is the first to demonstrate the power of these techniques for non-myopic Bayesian optimization.

## 2 The Two-Step Optimal (2-OPT) Acquisition Function

This section defines the two-step lookahead acquisition function. This acquisition function is optimal when there are two stages of measurements remaining, and so we call it 2-OPT. Before defining 2-OPT, we first provide notation and brief background from Gaussian process regression in Sect. 2.1. We then define 2-OPT in Sect. 2.2 and show how to estimate it with Monte Carlo in Sect. 2.3. While 2-OPT has been defined implicitly in past work, we include a complete description to provide a framework and notation supporting our novel efficient method for maximizing it in Sect. 3.

### 2.1 Gaussian process model for the objective $f$

We place a Gaussian process (GP) prior on the objective $f$. Although standard, here we briefly describe inference under a GP to provide notation used later. Our GP prior is characterized by a mean function $\mu(\cdot)$ and a kernel function $K(\cdot, \cdot)$. The posterior distribution on $f$ after observing $f$ at data points $D = (x^{(1)}, \ldots, x^{(m)})$ is a GP with mean function and kernel defined respectively by

$$
\mu(x) + K(x, D)K(D, D)^{-1}(f(D) - \mu(D)), \\
K(x, x') - K(x, D)K(D, D)^{-1}K(D, x').
\tag{1}
$$

In (1), $f(D) = (f(x^{(1)}), \ldots, f(x^{(m)}))$, and similarly for $\mu(D)$. Expressions $K(x, D)$, $K(D, x)$, and $K(D, D)$ similarly evaluate to a column vector, row vector, and square matrix respectively.

### 2.2 Two-step lookahead acquisition function

Here we define the 2-OPT acquisition function from a theoretical (but not yet computational) perspective. This formulation follows previous work on two-step and multi-step acquisition functions [Lam et al., 2016, Osborne et al., 2009, Ginsbourger and Riche, 2010, González et al., 2016]. 2-OPT gives optimal average-case behavior when we have two stages of evaluations remaining, and the second stage of evaluated may be chosen based on the results from the first.

To support batch evaluations while maintaining computational tractability, our first stage of evaluations uses a batch of $q \geq 1$ simultaneous evaluations, while the second stage uses a single evaluation.

Throughout, we assume that we have already observed a collection of data points $D$, so that the current posterior distribution is a GP with a mean function $\mu_0$ and kernel $K_0$ given by (1), and use $\mathbb{E}_0$ to indicate the expectation taken with respect to this distribution. We let $f_0^* = \min f(D)$ be the best point observed thus far.

We index quantities associated with the first stage of evaluations by 1 and the second by 2. We let $X_1$ indicate the set of $q$ points to be evaluated in the first stage. We let $f(X_1) = (f(x) : x \in X_1)$ indicate the corresponding vector of observed values and and let $\min f(X_1)$ be the smallest value in this vector. We let $x_2$ indicate the single point observed in the second stage.

For each $i = 1, 2$, we define $f_i^*$ to be smallest value observed by the end of stage $i$, so $f_1^* = \min(f_0^*, f(X_1))$ and $f_2^* = \min(f_1^*, f(x_2))$. We let $\mu_i$ be the mean function and $K_i$ the kernel for the posterior distribution given $D$ and observations available at the end of stage $i$. We let $\mathbb{E}_i$ indicate the expectation taken with respect to the corresponding Gaussian process.

The overall loss whose expected value we seek to minimize is $f_2^*$.

To find the optimal sampling strategy, we follow the dynamic programming principle. We first write the expected loss achievable at the end of the second stage, conditioned on the selection of points $(X_1)$ and results $(f(X_1))$ from the first stage. If we choose the final evaluation optimally, then this expected loss is $L_1 = \min_{x_2 \in \mathcal{A}} \mathbb{E}_1 [f_2^*]$. This posterior and thus also $L_1$ depends on $X_1$ and $f(X_1)$.

Following the derivation of the expected improvement Jones et al. [1998], we rewrite this as

$$L_1 = \min_{x_2 \in \mathcal{A}} \mathbb{E}_1 \left[ f_1^* - (f_1^* - f(x_2))^+ \right] = f_1^* - \max_{x_2 \in \mathcal{A}} \mathbb{E}_1 \left[ (f_1^* - f(x_2))^+ \right] = f_1^* - \max_{x_2 \in \mathcal{A}} \mathrm{EI}_1(x_2),$$

where $y^+ = \max(y, 0)$ is the positive part function and $\mathrm{EI}_1(x)$ is the expected improvement under the GP after the first evaluation has been performed:

$$\mathrm{EI}_1(x) = \mathrm{EI}(f_1^* - \mu_1(x_2), K_1(x_2, x_2)). \tag{2}$$

Here $\mathrm{EI}(m, v) = m\Phi(m/\sqrt{v}) + \sqrt{v}\varphi(m/\sqrt{v})$ gives the expected improvement at a point where the difference between the best observed point and the mean is $m$ and the variance is $v$. $\Phi$ is the standard normal cdf and $\varphi$ is the standard normal pdf.

With this expression for the value achievable at the start of the second stage, the expected value achieved at the start of the first stage is:

$$\mathbb{E}_0[L_1] = \mathbb{E}_0 \left[ f_1^* - \max_{x_2 \in \mathcal{A}} \mathrm{EI}_1(x_2) \right] = \mathbb{E}_0 \left[ f_0^* - (f_0^* - \min f(X_1))^+ - \max_{x_2 \in \mathcal{A}} \mathrm{EI}_1(x_2) \right]$$

$$= f_0^* - \mathrm{EI}_0(X_1) - \mathbb{E}_0 \left[ \max_{x_2 \in \mathcal{A}} \mathrm{EI}_1(x_2) \right], \tag{3}$$

where $\mathrm{EI}_0(X_1) = \mathbb{E}_0 \left[ (f_0^* - \min f(X_1))^+ \right]$ is the multipoints expected improvement [Ginsbourger et al., 2010] under the GP with mean $\mu_0$ and kernel $K_0$.

We define our two-step acquistion function to be

$$2\text{-}\mathrm{OPT}_\delta(X_1) = \mathrm{EI}_0(X_1) + \mathbb{E}_0 \left[ \max_{x_2 \in \mathcal{A}(\delta)} \mathrm{EI}_1(x_2) \right], \tag{4}$$

where $\mathcal{A}(\delta)$ is a set similar to $\mathcal{A}$ defined below. Because $f_0^*$ does not depend on $X_1$, finding the $X_1$ that minimizes (3) is equivalent to finding the value that maximizes (4) (when $\mathcal{A} = \mathcal{A}(\delta)$). In the definition of 2-OPT, we emphasize that $\mathbb{E}_0 \left[ \max_{x_2 \in \mathcal{A}(\delta)} \mathrm{EI}_1(x_2) \right]$ depends on $X_1$ through the fact that $f(X_1)$ influences the mean function $\mu_1$ and kernel $K_1$ from which $\mathrm{EI}_1$ is computed.

We define $\mathcal{A}(\delta)$ to be a compact set of points in $\mathcal{A}$ separated by at least $\delta$ from all points in $X_1$ and those with $K_0(x) = 0$. 2-OPT$(X_1)$ means 2-OPT$_\delta(X_1)$ with $\delta = 0$, i.e., with $\mathcal{A} = \mathcal{A}(\delta)$. The parameter $\delta \geq 0$ is introduced purely to overcome a technical hurdle in our theoretical result and we believe in practice it can be set to 0. Indeed, the theory allows setting $\delta$ to an extremely small value, such as $10^{-5}$, and the maximum of $\mathrm{EI}_1(x_2)$ over $x_2 \in \mathcal{A}$ is seldom this close to a point in $X_1$: the posterior variance vanishes at points in $X_1$ and $\mathrm{EI}_1(x_2)$ increases as $x_2$ moves away from them.

Figure 1 illustrates 2-OPT's behavior and shows how it explores more than EI.

### 2.3 Monte Carlo estimation of 2-OPT$(\cdot)$

2-OPT$_\delta(X_1)$ cannot be computed in closed form. We can, however, estimate it using Monte Carlo. We first use the reparameterization trick [Wilson et al., 2018] to write $f(X_1)$ as $\mu_0(X_1) + C_0(X_1)Z$, where $Z$ is a $q$-dimensional independent standard normal random variable and $C_0(X_1)$ is the Cholesky decomposition of $K_0(X_1, X_1)$.

We assume that $K_0(X_1, X_1)$ is positive definite so $C_0(X_1)$ is of full rank.

Then, under (1), for generic $x$,

$$\mu_1(x) = \mu_0(x) + K_0(x, X_1)K_0(X_1, X_1)^{-1}(f(X_1) - \mu_0(X_1))$$
$$= \mu_0(x) + K_0(x, X_1)(C_0(X_1)C_0(X_1)^T)^{-1}C_0(X_1)Z$$
$$= \mu_0(x) + \sigma_0(x, X_1)Z$$

$$K_1(x, x) = K_0(x) - K_0(x, X_1)K_0(X_1, X_1)^{-1}K(X_1, x)$$
$$= K_0(x) - K_0(x, X_1)(C_0(X_1)C_0(X_1)^T)^{-1}K(X_1, x)$$
$$= K_0(x) - \sigma_0(x, X_1)\sigma_0(x, X_1)^T.$$

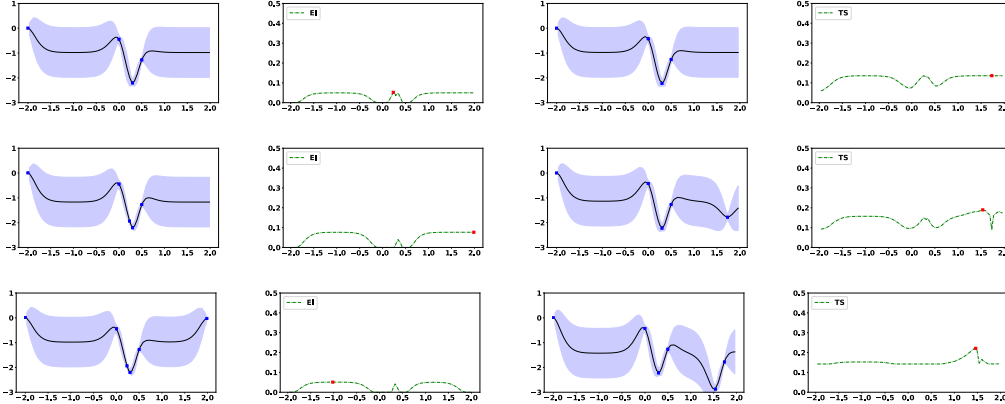

Figure 1: We demonstrate 2-OPT and EI minimizing a 1-d synthetic function sampled from a GP. Each row shows the posterior on $f$ (mean $+/-$ one standard deviation) and the corresponding acquisition function, for EI (left) and 2-OPT (right). We plot progress over three iterations. On the first iteration, EI evaluates a point that refines an existing local optimum and could have provided a small one-step improvement, but provides little information of use in future evaluations. In contrast, 2-OPT explores more aggressively, which helps it identify a new global minimum in the next iteration.

where $\sigma_0(x, X_1) = K_0(x, X_1)C_0(X_1)^{-1}$.

With this notation, we can write $\text{EI}_1(x_2)$ explicitly as

$$\text{EI}_1(x_2) = \text{EI}_1(X_1, x_2, Z) := \text{EI}(f_1^* - \mu_0(x_2) - \sigma_0(x_2, X_1)Z, K_0(x_2) - \sigma_0(x_2, X_1)\sigma_0(x_2, X_1)^T)$$

where we have introduced more explicitly in expanded notation $\text{EI}_1(X_1, x_2, Z)$ the quantities on which $\text{EI}_1(x_2)$ depends, and written it explicitly in terms of the function $\text{EI}(m, v)$.

Thus, we can rewrite the 2-OPT acquisition function as $\text{2-OPT}_\delta(X_1) = \mathbb{E}_0[\widehat{\text{2-OPT}}_\delta(X_1, Z)]$ where

$$\widehat{\text{2-OPT}}_\delta(X_1, Z) = (f_0^* - \min f(X_1))^+ + \max_{x_2 \in \mathcal{A}(\delta)} \text{EI}_1(x_2)$$

$$= \max(f_0^* - \mu_0(X_1) - C_0(X_1)Z)^+ + \max_{x_2 \in \mathcal{A}(\delta)} \text{EI}_1(X_1, x_2, Z),$$

where $\max(y)^+$ is the largest non-negative component of $y$, or 0 if all components are negative.

Then, to estimate $\text{2-OPT}_\delta(X_1)$, we sample $Z$ and compute $\widehat{\text{2-OPT}}_\delta(X_1, Z)$ using a nonlinear global optimization routine to calculate the inner maximization. Averaging many such replications provides a strongly consistent estimate of $\text{2-OPT}_\delta(X_1)$.

Previous approaches [Osborne et al., 2009, Ginsbourger and Riche, 2010, González et al., 2016] use this or a similar simulation method to obtain an estimator of 2-OPT, and then use this estimator within a derivative-free optimization approach. This requires extensive computation because:

1. The nested optimization over $x_2$ is time-consuming and must be done for each simulation.

2. Noise in the simulation requires either a noise-tolerant derivative-free optimization method that would typically require more iterations, or requires that the simulation be averaged over enough replications on each iteration to make noise negligible. This increases the number of simulations required to optimize accurately.

3. It does not leverage derivative information, causing optimization to require more iterations, especially as the dimension $d$ of the search space or the batch size $q$ grows.

## 3   Efficiently Optimizing 2-OPT

Here we describe a novel computational approach to optimizing 2-OPT that is substantially more efficient than previously proposed methods. Our approach includes two components: a novel simulation-based stochastic gradient estimator, which can be used within multistart stochastic gradient ascent; and variance reduction techniques that reduce the variance of this stochastic gradient estimator.

## 3.1 Estimation of the Gradient of 2-OPT

We now show how to obtain an approximately unbiased estimator of the gradient of $2\text{-OPT}_\delta(X_1)$. The main idea is to exchange the expectation and gradient operators when taking the gradient with respect to $X_1$,

$$\nabla 2\text{-OPT}_\delta(X_1) = \mathbb{E}_0 \left[ \nabla \widehat{2\text{-OPT}}_\delta(X_1, Z) \right]$$

$$= \mathbb{E}_0 \left[ \nabla \max(f_0^* - \mu_0(X_1) - C_0(X_1)Z)^+ + \nabla \max_{x_2 \in \mathcal{A}(\delta)} \text{EI}_1(X_1, x_2, Z) \right]$$

$$= \mathbb{E}_0 \left[ \nabla \max(f_0^* - \mu_0(X_1) - C_0(X_1)Z)^+ + \nabla \text{EI}_1(X_1, x_2^*, Z) \right]$$

where $x_2^* \in \arg\max_{x_2 \in \mathcal{A}(\delta)} \text{EI}_1(X_1, x_2, Z)$ is fixed and the last equation follows under some regularity conditions by the envelope theorem [Milgrom and Segal, 2002]. The following theorem shows this estimator of $\nabla 2\text{-OPT}_\delta(X_1)$ is unbiased. Its proof is in the supplement.

**Theorem 1.** *We assume:*

- *The domain $\mathcal{A}(\delta)$ is nonempty and compact and $\delta > 0$.*

- *The mean function $\mu_0$ and kernel $K_0$ are continuously differentiable.*

- *The kernel $K_0$ is non-degenerate, in the sense that the posterior variance, $K_1(x, x)$, at a point is non-zero if the prior variance, $K_0(x, x)$, is strictly positive and that point has not been sampled ($x$ is not in $X_1$).*

*Let $x_2^*$ be a global maximizer in $\mathcal{A}(\delta)$ of $\text{EI}_1(X_1, x_2, Z)$. Then,*

$$g(X_1, Z) := \nabla \max(f_0^* - \mu_0(X_1) - C_0(X_1)Z)^+ + \nabla \text{EI}_1(X_1, x_2^*, Z) \tag{5}$$

*exists almost surely and is an unbiased estimator of $\nabla 2\text{-OPT}_\delta(X_1)$, where the gradient is taken with respect to $X_1$ while holding $A(\delta)$ fixed.*

We then use this stochastic gradient estimator within stochastic gradient ascent [Kushner and Yin, 2003] with multiple restarts to find a collection of stationary points $X_1$ (each $X_1$ is a single point in $\mathbb{R}^d$ if $q = 1$ or a collection of $q$ points in $\mathbb{R}^d$ if $q > 1$). We use Monte Carlo to evaluate $2\text{-OPT}(X_1)$ for each of these stationary points and select as our approximate maximizer of 2-OPT the point or batch of points with the largest estimated $2\text{-OPT}(X_1)$. In practice we perform this procedure using $\delta = 0$, although Theorem 1 only guarantees an unbiased gradient estimator when $\delta > 0$.

## 3.2 Variance reduction

We now describe variance reduction techniques that further improve computation time and accuracy.

**Gauss-Hermite Quadrature (fully sequential setting)**   In the fully sequential setting where we propose one point at each iteration ($q = 1$), we use Gauss-Hermite quadrature [Liu and Pierce, 1994] to estimate $2\text{-OPT}(X_1)$ and its gradient. These quantities are both expectations over the 1-d standard Gaussian random variable $Z$. Gauss-Hermite quadrature estimates the expectation of a random variable $g(Z)$ by a weighted sum $\sum_{i=1}^n w_i g(z_i)$ with well-chosen weights $w_i$ and locations $z_i$. In practice, we find $n = 20$ accurately estimates $2\text{-OPT}(X_1)$ and its gradient.

**Importance sampling (batch setting)**   In the batch setting, Gauss-Hermite quadrature scales poorly with batch size $q$ since the number of weighted points required grows exponentially with the dimension over which we integrate, which is $q$. In the batch setting, we adopt another variance reduction technique: importance sampling [Asmussen and Glynn, 2007].

Recall that our Monte Carlo estimator of 2-OPT and its gradient involve a sampled multipoints EI term $\max(f_0^* - \mu_0(X_1) - C_0(X_1)Z)^+$.

For high-dimensional test functions or after we have many function evaluations, most draws of $Z$ result in this multipoints EI term taking a value of $0$. This occurs when all components of $\mu_0(X_1) + C_0(X_1)Z$ are larger than $f_0^*$. For such $Z$, the derivative of this immediate improvement

term is also $0$. Also, for such $Z$, the second term in our Monte Carlo estimator of 2-OPT and its gradient, $\max_{x_2 \in \mathcal{A}} \text{EI}_1(X_1, x_2, Z)$, also tend to be small and have a small gradient.

As a result, when calculating the expected value of these samples of 2-OPT or its gradient, we include many 0s. This can make the variance of estimators based on averaging these estimators large relative to their expected value. This in turn makes gradient-based optimization and comparison using Monte Carlo estimates challenging.

To address this, we simulate $Z$ from a multivariate Gaussian distribution with a larger standard deviation $v > 1$, calling it $Z^v$. This substantially increases the chance that the at least one component of $\mu_0(X_1) + C_0(X_1)Z$ will exceed $f_0^*$. We find $v = 3$ works well in test problems.

To compensate for sampling from a different distribution, we multiply by the likelihood ratio between the density for which we wish to calculate the expectation, which is the multivariate standard normal density, and the density from which $Z^v$ was sampled. Letting $\varphi(\cdot; 0, v^2 I)$ indicate the $q$-dimensional normal multivariate density with mean 0 and covariance matrix $v^2 I$, this likelihood ratio is $\varphi(Z^v; 0, I)/\varphi(Z^v; 0, v^2 I)$.

The resulting unbiased estimators of 2-OPT and its gradients are, respectively, $\widehat{\text{2-OPT}}(X_1, Z^v)\varphi(Z^v; 0, I)/\varphi(Z^v; 0, v^2 I)$ and $g(X_1, Z^v)\varphi(Z^v; 0, I)/\varphi(Z^v; 0, v^2 I)$.

# 4 Numerical experiments

We test our algorithms on common synthetic functions and widely-benchmarked real-world problems. We compare with acquisition functions widely used in practice including GP-LCB [Srinivas et al., 2010], PI, EI [Snoek et al., 2012] and KG [Wu and Frazier, 2016] and multi-step lookahead methods from Lam et al. [2016] and González et al. [2016].

2-OPT is substantially more robust than competing methods, providing performance that is best or close to best across essentially all problems, iterations and performance measures. In contrast, while other methods like EI and KG sometimes outperform 2-OPT, they also sometimes substantially underperform. For example, EI has simple regret two orders of magnitude worse than 2-OPT on Hartmann6 and KG is 3 times worse on 10d Rosenbrock.

Moreover, the computation time of one iteration of 2-OPT is fast enough to be practical, varying from seconds to several minutes on a single core in all our experiments, and can be easily parallelized across cores. This is approximately an order of magnitude faster than the benchmark multi-step lookahead methods. 2-OPT's strong empirical performance together with a supporting fast computational method unlocks the value of two-step lookahead in practice.

**Experimental details**   Following Snoek et al. [2012], we use a constant mean prior and the ARD Matérn $5/2$ kernel. We integrate over GP hyperparameters by sampling 16 sets of values using the `emcee` package [Foreman-Mackey et al., 2013]. We initiate our algorithms by randomly sampling 3 points from a Latin hypercube design and then start the Bayesian optimization iterative process. We use 100 random initializations in the synthetic and real functions experiments, 40 in the comparisons to multi-step lookahead methods (replicating the experiment setup of Lam et al. [2016]), and 10 for comparisons of computation time.

**Synthetic functions, compared with one-step methods.**   First, we test 2-OPT and benchmark methods on 6 well-known synthetic test functions chosen from Bingham [2015] ranging from 2d to 10d: 2d Branin, 2d Camel, 5d Ackley5, 6d Hartmann6, 8d Cosine and 10d Levy. Figure 2 shows the 90% quantile of the log immediate regret for 6 of these 8 benchmarks. Figure 5 in the supplement reports the mean of the base 10 logarithm of the immediate regret (plus or minus one standard error) on these functions along with two more added in the author response period: 2d Michalewicz and 10d Rosenbrock.

**Synthetic functions, compared with multi-step methods.**   To compare with non-myopic algorithms proposed in González et al. [2016] and Lam et al. [2016], we replicate the experimental settings in Lam et al. [2016] and add 2-OPT's performance to their Table 2. We report the results in Table 1. GLASSES was proposed in González et al. [2016] and the four columns R-4-9, R-4-10, R-5-9, and R-5-10 are algorithm variants proposed in Lam et al. [2016].

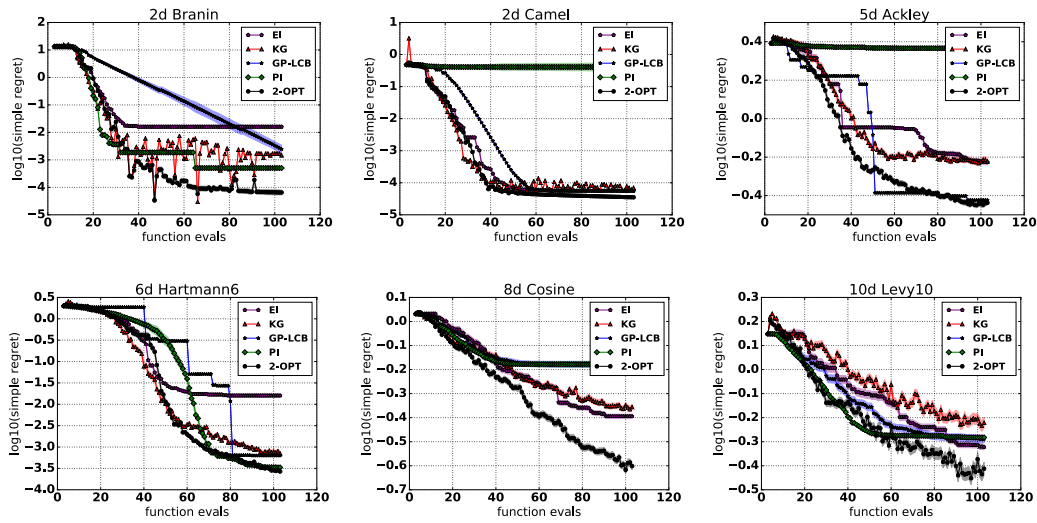

Figure 2: Synthetic test functions, 90% quantile of log10 immediate regret compared with common one-step heuristics. 2-OPT provides substantially more robust performance.

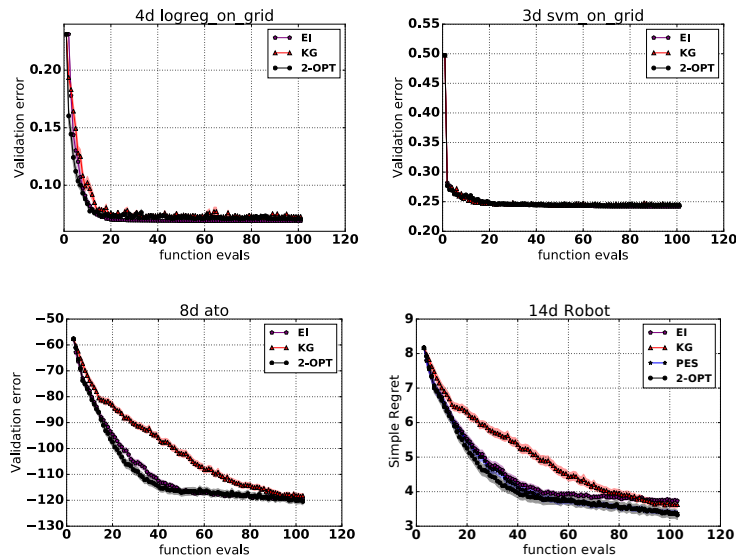

Figure 3: Realistic benchmarks: HPOlib (top): 2-OPT is competitive with the best of the competitors in each benchmark. ATO (bottom left): 2-OPT outperforms EI slightly and clearly outperforms KG. All algorithms converge to nearly the same performance. Robot Pushing: 2-OPT slightly outperforms PES and clearly outperforms EI and KG.

| Function name | | PI | EI | UCB | PES | GLASSES | R-4-9 | R-4-10 | R-5-9 | R-5-10 | 2-OPT |
|---|---|---|---|---|---|---|---|---|---|---|---|
| Branin-Hoo | Mean | .847 | .818 | .848 | .861 | .846 | .904 | .898 | .887 | .903 | **.9995** |
| | Median | .922 | .909 | .910 | .983 | .909 | .959 | .943 | .921 | .950 | **.9994** |
| Goldstein-Price | Mean | .873 | .866 | .733 | .819 | .782 | .895 | .784 | .861 | .743 | **.9651** |
| | Median | .983 | .981 | .899 | .987 | .919 | **.991** | .985 | .989 | .928 | **.9911** |
| Griewank | Mean | .827 | .884 | .913 | **.972** | 1.0[2] | .882 | .885 | .930 | .867 | .9321 |
| | Median | .904 | .953 | .970 | **.987** | 1.0[2] | .967 | .962 | .960 | .954 | .9801 |
| 6-hump Camel | Mean | .850 | .887 | .817 | .664 | .776 | .860 | .825 | .793 | .803 | **.9010** |
| | Median | .893 | **.970** | .915 | .801 | .941 | .926 | .900 | .941 | .907 | .9651 |

Table 1: Performance of our two-step acquisition fuction (2-OPT) on test functions compared with non-myopic and other benchmark algorithms originally reported in Lam et al. [2016]. Each value reported is the "gap": the ratio of the overall improvement obtained by the algorithm to the improvement possible by a globally optimal solution. A gap of 1 represents finding the optimal solution; 0 represents no improvement in solution quality. The best gap appears in boldface.

Values reported are the "gap" [Huang et al., 2006], which is the ratio of the improvement obtained by an algorithm to the improvement possible by a globally optimal solution. Letting $\hat{x}_N$ be the best solution found by the algorithm and $\hat{x}_0$ be the best solution found in the initial stage, the gap is $G = (f(\hat{x}_0) - f(\hat{x}_N))/(f(\hat{x}_0) - \min_x f(x))$. A gap of 1 indicates that the algorithm found a globally optimal solution, while 0 indicates no improvement.

2-OPT is best in 5 out of 8 problems (tied for best on one of these problems), and second-best in the remaining 3. It outperforms or ties the non-myopic competitiors on all problem instances.

In the supplement Figure 4 shows the time required for acquisition function optimization on 1 core from a AWS t2.2xlarge instance for 2-OPT, EI, KG, and GLASSES. Time for other problems is similar, with higher-dimensional problems requiring more time. 2-OPT's computation time is comparable to KG, about 10 times slower than EI, and about 10 times faster than GLASSES. Code from Lam et al. [2016] was unavailable when these experiments were performed, but Lam [2018] reports that the time required is between 10 minutes and 1 hour, even on low-dimensional problems.

**Realistic benchmarks**   Figure 3 shows performance on a collection of more realistic benchmarks, HPOlib, ATO, and Robot Pushing.

The HPOlib library was developed in Eggensperger et al. [2013] based on hyperparameter tuning benchmarks from Snoek et al. [2012]. We benchmark on the two most widely used test problems there: logistic regression and SVM. On both problems, 2-OPT performs comparably to the best of the competitors, with 2-OPT and EI slightly outperforming KG on logistic regression.

The assemble-to-order (ATO) benchmark [Hong and Nelson, 2006, Poloczek et al., 2017] is a reinforcement learning problem with a parameterized control policy where the goal is to optimize an 8-dimensional inventory target vector to maximize profit in a business setting. 2-OPT provides a substantial benefit over competitors from the start and remains best over the whole process. After 40 iterations, EI catches 2-OPT, while KG lags both EI and 2-OPT until iteration ~100 where all the algorithms converge with comparable performance.

The robot pushing problem is a 14-dimensional reinforcement learning problem considered in Wang and Jegelka [2017]. 2-OPT outperforms all the competitors on this benchmark.

## 5   Conclusions

In this article, we propose the first computationally efficient two-step lookahead BayesOpt algorithm. The algorithm comes in both sequential and batch forms, and reduces the computational time compared to previous proposals with increased performance. In experiments, we find that two-step lookahead provides additional value compared to several one-step lookahead heuristics.

## Footnotes

[2]Lam et al. [2016] reports that GLASSES achieves gap=1 on Griewank because it arbitrarily evaluates at the origin, which happens to be a global minimizer. Following Lam et al. [2016], we exclude these results.

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
