[Supplementary Material · two-step-supplement.pdf]

# Practical Two-Step Look-ahead
# Bayesian Optimization

*Supplementary Material*

## A   Proof of Theorem 1

To prove Theorem 1, we need to prove the interchange of the expectation and the gradient operators is valid.

We fix $X_1$ and $\mathcal{A}(\delta)$. We then choose $i \in [q] = \{1, \ldots, q\}$ representing a point within the first stage of points $X_1$ and a component $j \in [d] = \{1, \ldots, d\}$ of that point. For real-valued $\epsilon$, we then let $X_1(\epsilon)$ be $X_1$ but with this component replaced by its sum with $\epsilon$. Then,

$$\widehat{\text{2-OPT}}_\delta(X_1(\epsilon), Z) = \max(f_0^* - \mu_0(X_1(\epsilon)) - C_0(X_1(\epsilon))Z)^+ + \text{EI}_1(X_1(\epsilon), x_2^*(\epsilon, Z), Z),$$

We then choose an open set $\Theta \subset \mathbb{R}$ containing 0 such that $K_0(X_1(\epsilon), X_1(\epsilon))$ and hence $C_0(X_1(\epsilon))$ is (strictly) positive definite for each $\epsilon \in \Theta$. This is possible because $K_0(X_1(0), X_1(0))$ was assumed positive definite. We also choose $\Theta$ with the requirement that $\sup_{\epsilon \in \Theta} |\epsilon| \le \delta/2$.

Here, we have modified our notation to $x_2^*(\epsilon, Z) \in \operatorname{argmin}_{x_2 \in \mathcal{A}} \text{EI}_1(X_1(\epsilon), x_2, Z)$ (called $x_2^*$ in the body of the paper) to note dependence on $\epsilon$ and $Z$.

With this notation, the claimed validity of this interchange can be restated as the claim that

$$\frac{\partial}{\partial \epsilon} \text{2-OPT}_\delta(X_1(\epsilon)) = \mathbb{E}_0 \left[ \frac{\partial}{\partial \epsilon} \widehat{\text{2-OPT}}_\delta(X_1(\epsilon), Z) \right] \tag{6}$$

To prove that (6) is valid, we use Theorem 1 in L'Ecuyer [1990]. This theorem requires three sufficient conditions:

- (i) $\widehat{\text{2-OPT}}_\delta(X_1(\epsilon), Z)$ is continuous in $\epsilon$ over $\Theta$ for any fixed $Z$;
- (ii) $\widehat{\text{2-OPT}}_\delta(X_1(\epsilon), Z)$ is differentiable in $\epsilon$ except on a denumerable set in $\Theta$ for any given $Z$;
- (iii) the derivative of $\widehat{\text{2-OPT}}_\delta(X_1(\epsilon), Z)$ with respect to $\epsilon$ (when it exists) is uniformly bounded by a random variable $M(Z)$ for all $\epsilon \in \Theta$ and the expectation of $M(Z)$ is finite.

Before proving these conditions, we first state several lemmas.

**Lemma 1.** $\text{EI}(m, v) = m\Phi(m/\sqrt{v}) + \sqrt{v}\varphi(m/\sqrt{v})$ *is continuously differentiable in* $m, v$ *for any* $m \in \mathbb{R}$ *and any* $v$ *in* $(0, \infty)$.

*Proof.* The following expressions can be verified from direct differentiation, and also appear in slightly modified form in Jones et al. [1998]:

$$\frac{\partial}{\partial \sqrt{v}} \text{EI}(m, v) = \varphi\left(\frac{m}{\sqrt{v}}\right), \quad \frac{\partial}{\partial m} \text{EI}(m, v) = \Phi\left(\frac{m}{\sqrt{v}}\right)$$

The chain rule then implies

$$\frac{\partial}{\partial v} \text{EI}(m, v) = \frac{1}{2\sqrt{v}} \varphi\left(\frac{m}{\sqrt{v}}\right)$$

These expressions for $\frac{\partial}{\partial m} \text{EI}(m, v)$ and $\frac{\partial}{\partial v} \text{EI}(m, v)$ are continuous in $m$ and $v$ over the claimed ranges, which notably exclude $v = 0$. $\square$

**Lemma 2.** $K_1(x_2, \epsilon) := K_0(x_2) - K_0(x_2, X_1(\epsilon))K_0(X_1(\epsilon))^{-1}K_0(X_1(\epsilon), x_2)$ *and* $\sigma_0(x_2, X_1(\epsilon))$ *are continuously differentiable in* $x_2$ *and* $\epsilon$ *for all* $\epsilon \in \Theta$ *and all* $x_2 \in \mathcal{A}(\delta)$.

*Proof.* Recall $\sigma_0(x_2, X_1(\epsilon)) = K_0(x_2, X_1(\epsilon))C_0(X_1(\epsilon))^{-1}$ and $C_0(X_1(\epsilon))$ is the Cholesky decomposition of $K_0(X_1(\epsilon))$.

$K_0(X_1(\epsilon))$ is positive definite for all $\epsilon \in \Theta$ (we chose $\Theta$ so this would be true). Also, (1) the Cholesky decomposition is continuously differentiable Smith [1995]; (2) the matrix inverse is continuously differentiable for positive definite matrices; and (3) $K_0$ is continuously differentiable. The result follows since compositions, products, and sums of continuously differentiable functions are continuously differentiable. $\qquad\square$

**Lemma 3.** *$K_1(x_2, \epsilon)$ is bounded below by a strictly positive constant $r > 0$ across all $x_2 \in \mathcal{A}(\delta)$ and all $\epsilon \in \Theta$.*

*Proof.* All points in $\mathcal{A}(\delta)$ have $K_0(x_2) > 0$. Also, since $\epsilon \leq \delta/2$ for all $\epsilon \in \Theta$ (we chose $\Theta$ so this would be true), all points in $\mathcal{A}(\delta)$ are separated from all points in $X_1(\epsilon)$ by at least $\delta - \epsilon \geq \delta/2 > 0$. Thus, by the assumption that the kernel is non-degenerate in the statement of the theorem, the posterior variance $K_1(x_2, \epsilon)$ is strictly positive for all $x_2 \in \mathcal{A}(\delta)$.

Also, $K_1(x_2, X_1(\epsilon))$ is continuous by Lemma 2. Thus, since $\mathcal{A}(\delta)$ is compact, the infimum over $x_2 \in \mathcal{A}(\delta)$ is attained within $\mathcal{A}(\delta)$. This infimum is thus strictly positive. $\qquad\square$

**Lemma 4.** *Consider any fixed $Z$. Then, $\max_{x_2 \in \mathcal{A}(\delta)} \mathrm{EI}_1(X_1(\epsilon), x_2, Z)$ is differentiable for almost every $\epsilon \in \Theta$. At each $\epsilon_0$ for which this derivative exists, the derivative is equal to*

$$\frac{d}{d\epsilon} \mathrm{EI}_1(X_1(\epsilon), x_2^*(\epsilon_0, Z), Z), \tag{7}$$

*where either $x_2^*(\epsilon_0, Z) \in \mathrm{argmin}_{x_2 \in \mathcal{A}(\delta)} \mathrm{EI}_1(X_1(\epsilon_0), x_2, Z)$ is unique or (7) does not depend on the choice within this set.*

*Proof.* To show this result, the envelope theorem (Corollary 4 of Milgrom and Segal 2002) tells us that it is sufficient to verify the following conditions:

1. $\mathcal{A}(\delta)$ is a non-empty compact space;

2. $\mathrm{EI}_1(X_1(\epsilon), x_2, Z)$ is continuous in $x_2$;

3. $\frac{d}{d\epsilon} \mathrm{EI}_1(X_1(\epsilon), x_2, Z)$ is continuous in $\epsilon$ and $x_2$.

This will then imply absolute continuity of $\max_{x_2 \in \mathcal{A}(\delta)} \mathrm{EI}_1(X_1(\epsilon), x_2, Z)$ (implying differentiability for almost every $\epsilon$) and the claimed expression for the derivative.

The first condition is assumed in the statement of Theorem 1.

We now verify the second and third conditions. Recall that

$$\mathrm{EI}_1(X_1(\epsilon), x_2, Z) = \mathrm{EI}(f_1^* - \mu_0(x_2) - \sigma_0(x_2, X_1(\epsilon))Z, K_1(x_2, \epsilon))$$

The second condition follows from continuity of EI (Lemma 1), $\mu_0$ (assumed in the statement of the Theorem), $\sigma_0(x_2, X_1(\epsilon))$ (Lemma 2), and $K_1(x_2, \epsilon)$ (Lemma 2).

The third condition follows from the fact that $K_1(x_2, \epsilon)$ stays bounded away from 0 (Lemma 3), $\mathrm{EI}(m, v)$ is continuously differentiable when $v > 0$ (Lemma 1), continuous differentiability of $\mu_0(x_2)$ (assumed in the statement of the Theorem), and continuous differentiability of $\sigma_0(x_2, X_1(\epsilon))$ and $K_1(x_2, \epsilon)$ (Lemma 2). $\qquad\square$

With these lemmas, we now proceed to show the conditions required by L'Ecuyer [1990].

## A.1 Proof of condition (i)

Because the the mean function $\mu_0$ and the kernel $K_0$ are assumed continuous, $\mu_0(X_1(\epsilon))$ and $C_0(X_1(\epsilon))$ are continuous in $\epsilon$.

Since the maximum of several continuous functions is continuous, $\max(f_0^* - \mu_0(X_1(\epsilon)) - C_0(X_1(\epsilon))Z)^+$ is continuous in $\epsilon$.

Continuity of $\mathrm{EI}_1(X_1(\epsilon), x_2^*(\epsilon, Z), Z)$ was shown in Lemma 4.

Since the sum of continuous functions is continuous, $\widehat{\text{2-OPT}}_\delta(X_1(\epsilon), Z)$ is continuous in $\epsilon$.

## A.2 Proof of condition (ii)

Fix any $Z$. Leveraging Lemma 4, it is sufficient to show that $\max(f_0^* - \mu_0(X_1(\epsilon)) - C_0(X_1(\epsilon))Z)^+ +$ $\mathrm{EI}_1(X_1(\epsilon), x_2^*, Z)$ is differentiable with respect to $\epsilon$ except on a denumerable set in $\Theta$.

Let $D \subseteq \Theta$ be the set of values of $\epsilon$ such that $\max(f_0^* - \mu_0(X_1(\epsilon)) - C_0(X_1(\epsilon))Z)^+$ is not differentiable. We have

$$D \subset \cup_{i,j \in 0:q} \left\{ \epsilon \in \Theta : h_i(\epsilon) = h_j(\epsilon), \frac{dh_\epsilon(i)}{d\epsilon} \neq \frac{dh_j(\epsilon)}{d\epsilon} \right\}$$

where $h_0(\epsilon) = 0$ and $h_i(\epsilon)$ for $i > 0$ is a component $i$ of $f_0^* - \mu_0(X_1(\epsilon)) - C_0(X_1(\epsilon))Z$. Thus it is sufficient to show that

$$\left\{ \epsilon \in \Theta : h_i(\epsilon) = h_j(\epsilon), \frac{dh_\epsilon(i)}{d\epsilon} \neq \frac{dh_j(\epsilon)}{d\epsilon} \right\}$$

is denumerable.

Define $\eta(\epsilon) := h_i(\epsilon) - h_j(\epsilon)$. Observe that differentiability of $\mu_0$ and $K_0$ imply differentiability of $\eta$. We would like to show that $E := \left\{ \epsilon \in \Theta : \eta(\epsilon) = 0, \frac{d\eta(\epsilon)}{d\epsilon} \neq 0 \right\}$ is denumerable. To prove this, it is sufficient to show that $E$ contains only isolated points because any set of isolated points in $\mathbb{R}$ is denumerable (see the proof of statement 4.2.25 on page 165 in Thomson et al. [2008]).

We prove that $E$ only contains isolated points by contradiction. Suppose that $\epsilon_* \in E$ is not an isolated point. Then, there is a sequence of points $\epsilon_1, \epsilon_2, \ldots$ in $E$ that converge to $\epsilon_*$. Then, noting that $\eta(\epsilon_n) = \eta(\epsilon) = 0$, we have

$$0 \neq \frac{d\eta(\epsilon)}{d\epsilon}\Big|_{\epsilon=\epsilon_*} = \lim_{n \to \infty} \frac{\eta(\epsilon_n) - \eta(\epsilon_*)}{\epsilon_n - \epsilon_*} = \lim_{n \to \infty} 0 = 0,$$

which is a contradiction. Thus we may conclude that $E$ only contains isolated points, and so is denumerable.

## A.3 Proof of condition (iii)

We first prove that $\frac{\partial}{\partial \epsilon} \max(f_0^* - \mu_0(X_1(\epsilon)) - C_0(X_1(\epsilon))Z)^+$, when it exists, has a magnitude bounded above by

$$\left| \frac{\partial}{\partial \epsilon} \max(f_0^* - \mu_0(X_1(\epsilon)) - C_0(X_1(\epsilon))Z)^+ \right| \leq M_1 + M_2 \sum_i |Z_i|$$

where $M_1$ is the maximum of the absolute value of the derivatives of the components of $\mu_0(X_1)$ with respect to $\epsilon$ and, similarly, $M_2$ is the maximum of the absolute value of the derivative of the entries of $C_0(X_1)$ with respect to $\epsilon$. Because $\mu_0$ and $K_0$ are both assumed continuously differentiable, $M_1$ and $M_2$ are finite. We then have that $\mathbb{E}[M_1 + M_2 \sum_i |Z_i|]$ is finite.

We now concentrate on the second term in $\widehat{\text{2-OPT}}_\delta(X_1(\epsilon), Z)$. By Lemma 4, when it exists, $\frac{\partial}{\partial \epsilon} \max_{x_2 \in \mathcal{A}} \mathrm{EI}_1(X_1(\epsilon), x_2, Z)|_{\epsilon=\epsilon_0} = \frac{\partial}{\partial \epsilon} \mathrm{EI}_1(X_1(\epsilon), x_2^*(\epsilon_0, Z), Z)|_{\epsilon=\epsilon_0}$.
Recall that

$$\mathrm{EI}_1(X_1(\epsilon), x_2, Z) = \mathrm{EI}(\mu_0(x_2) + \sigma_0(x_2, X_1(\epsilon))Z, K_0(x_2) - \sigma_0(x_2, X_1(\epsilon))\sigma_0(x_2, X_1(\epsilon))^T).$$

We will bound the derivative of this quantity with respect to $\epsilon$ by a constant.

In the proof of Lemma 4, we showed that $\frac{\partial}{\partial \epsilon} \sigma_0(x_2, X_1(\epsilon))$ is continuous in $x_2$ and $\epsilon$, and so its components are bounded over $\Theta$ (since we assumed $\Theta$ is contained in a compact set). This bound does not depend on $Z$. Call this constant $M_3$.

We then use the chain rule to provide an expression for $\frac{\partial}{\partial \epsilon} \mathrm{EI}_1(X_1(\epsilon), x_2, Z)$. Recalling that $\mathrm{EI}_1(X_1(\epsilon), x_2, Z)$ can be written more explicitly as $\mathrm{EI}(f_1^* - \mu_0(x_2) - \sigma_0(x_2, X_1)Z, K_1(x_2, \epsilon))$ we first note that the partial derivatives of EI with respect to its first and second arguments are non-negative (provided in Lemma 1) and can be bounded above by 1 and $\varphi(0)/2\sqrt{r}$ respectively (leveraging Lemma 3). The derivative of the first argument with respect to $\epsilon$ is the sum of:

- the derivative of $f_1^* = \min(f_0^*, \min \mu_0(X_1(\epsilon) + C_0(X_1(\epsilon))Z)$, whose absolute value is bounded by the largest component of $\frac{\partial}{\partial \epsilon}\mu_0(X_1(\epsilon)) + C_0(X_1(\epsilon))Z$;

- $\frac{\partial}{\partial \epsilon}\sigma_0(x_2, X_1(\epsilon))Z$.

Since $\mu_0, C_0$, and $\sigma_0$ are all continuously differentiable in $\epsilon$, $\Theta$ is contained within a compact set, and the maximum of a continuous function over a compact set is finite, the magnitude of these quantities can all be bounded above by a finite constant times $|Z|$.

The derivative of the second argument is continuous in $\epsilon$ (Lemma 2) and so has a maximum that is similarly bounded above by a constant over $\Theta$.

Thus, $|\frac{\partial}{\partial \epsilon}\mathrm{EI}_1(X_1(\epsilon), x_2, Z)|$ is bounded above by a linear function $|Z|$, and a linear function of $|Z|$ is integrable.

## B  Additional Experiments

Here we include plots of numerical experiments discussed in the main paper, but that could not be included there due to space constraints. Figure 4 shows computation time compared with EI, KG, and GLASSES. Figure 5 shows mean performance across a collection of 8 widely used synthetic benchmarks against common one-step heuristics.

Figure 4: Run time benchmarks: 2-OPT is clear better than GLASSES and comparable to popular one-step heuristics.

Figure 5: Benchmarks of 2-OPT with common one-step heuristics: EI, PI, KG and GP-LCB on eight common synthetic functions. 2-OPT outperforms the competitors on 7 out of 8 test functions, although some of the one-step algorithms are known to be highly effective on these functions.