[Reviews · NeurIPS 2019]

Reviewer 1



General comments / questions (see elsewhere in this review for contributions/originality/significance): - The paper is generally well-written and clear given the (at times) technical content. - The approach appears technical sound and I find the insight presented in section 3.1 (and proof in the supplementary) together with resulting algorithm quite interesting (and possibly useful in other domains). I’ve read through and haven’t detected any obvious flaws but I’d suggest to double check the technical proofs and assumptions (especially in the appendix). - Sec 2 seems a bit long compared to the new insight presented in Sec 3. - The experiments appear relevant and mostly well-executed. My main concern is the perceived benefit of the two-step approach. It is especially difficult for a reader to quickly see the clear benefit of the 2-OPT (or any variant of it) when looking at the “real-world” HPOlib and ATO benchmarks. It would perhaps be relevant to further focus on and emphasise the robustness of the 2-step approach over alternatives. Also, given that a primary goal of the paper is to improve time complexity, not introduce a new acquisition function per se, I would perhaps have expected an more detailed empirical comparison of the time complexity among different multi-step (at least two) BO approaches (especially GLASSES) (possibly in the appendix). - The variance on the optimization traces for EI and LCB are often very small (despite a very non-smooth traces for e.g. 5d Ackley fig 2) or non-existent; any reason or insight into this? Minor: - L 143: Define Q (I assume it is a typo or last-minute change in notation). - L 233-244; please check for minor typos; esp. l 238 - Figures are bit hard to read due to axis labels and markers etc being rather small; overlapping graphs etc.

Reviewer 2



Summary: This paper proposes a new acquisition function for Bayesian optimization that considers not only one step in the future, but two. This is expected to be more accurate. However, taking into account that there are 2 evaluations remaining is a challenging task. The required computations are intractable and approximations need to be made. The authors suggest using a Monte Carlo approximation of the objective, which is then combined with stochastic gradient optimization. Some theoretical results about the convergence of the proposed approach are given. The method is also evaluated and compared with other non-myopic approaches for BO alongside with standard myopic approaches. These experiments show some gains. Comments: I believe that the problem addressed by the paper is interesting and relevant for the community. The problem is, however, that the proposed approach seems to be a slight variation of the already known methods. In particular, the only contribution seems to be the optimization of the acquisition function which is done using stochastic gradient algorithms. This seems to be a rather weak contribution. The experimental setting seems to be limited in the sense that only a few real datasets are provided, and in those (HPOlib) the proposed approach does not seem to give significantly better results. The authors only compare with related methods only on a set of synthetic problems extracted from another reference. I wonder why the did not compare on each problem with those techniques. The authors also claim that the proposed approach is faster than competing methods. However, no experiments showing this are provided. The authors also say that their approach supports using a batch evaluation setting for the first step. However, in the experiments, it is not clear what batch size they use. There are also some typos in the paper: E.g. line 110, line 125, line 143 (what is Q?), line 154 (what is EI_2?), and line 152 (K(X_1,x) is missing a subindex). From my point of view the paper rather weak and marginally below the acceptance threshold. The reproducibility checklist is wrongly completed by the authors. Summing up, I believe that this paper needs more work. In particular, the experimental section needs to be improved and to actually show the benefits of the proposed approach.

Reviewer 3



Originality: This problem has been studied before. However, this paper proposes a method for estimating the gradients of the acquisition functions. Quality: The paper has some theoretical analysis. However, more experimental results are required. Clarity: The paper can be understood. However, some of the text requires rewriting to make it easier to understand. Significance: With more empirical results, more researchers can use the proposed method.

[Author Response · NeurIPS 2019]

Benefit of the two-step approach (R1, R2): As R1 suggested, we examined 2-OPT's robustness as measured by the 90%
quantile of the (log) regret over replications (see figure). By this measure, 2-OPT improves more significantly in each
problem over the best competing method. 2-OPT is also more robust than competing methods when looking at the
mean regret (or 90% quantile) across *problems*. 2-OPT is better than or comparable to the best other method in nearly
all problems and iterations. In contrast, consider EI. By mean performance, EI performs as well as 2-OPT on several
problems (logistiic regression, SVM, ATO, and 2d Camel) but underperforms in others (Ackley, Cosine, Levy) and
sometimes severely so (Branin, Hartmann6 — note that performance is on a log scale). Other methods behave similarly,
performing as well as 2-OPT in some problems but underperforming significantly in others.

Computational Cost (R1, R2, R3): The figure below shows the time required for acquisition function optimization on a
single core using AWS instances. Time for other problems is similar, with higher-dimensional problems requiring more
time. 2-OPT's computation time is comparable to KG, about 10 times slower than EI, and about 10 times faster than
GLASSES. (Code from Lam et al. 2016 is unavailable.) We'll report computation time in the final version.

While 2-OPT's computation per iteration can be substantial, the multiple (1000) restarts of SGD used by 2-OPT can be
trivially parallelized with a linear speedup. Parallelizing all starts gives $< 2$ seconds of wall-clock time per iteration
for all problems. Moreover, for objective functions that require several hours per evaluation on a multi-core machine,
being able to find a good solution with fewer time-consuming objective function evaluations often merits the additional
overhead required to optimize a more sophisticated acquisition function. Since 2-OPT has more robust query efficiency
than other methods, and is as fast as KG and GLASSES, we feel that 2-OPT is of significant practical value in an
important range of problems: those objective function evaluation are costly enough to make improving query efficiency
over EI and other faster myopic methods worth the additional computational cost.

More experimental settings (R2, R3): We ran experiments for Rosenbrock (see figure) and will add Michalwicz and
Robot pushing to the final version. Goldstein-Price is available in Table 1.

"Authors compare with related [non-myopic] methods only on a set of synthetic problems extracted from another
reference." (R2) At submission we lacked code for GLASSES and Lam et al. 2016. We recently obtained code for
GLASSES and will include comparisons in the final version. We're awaiting email replies from Lam et al. If we obtain
code in time we'll also include those comparisons in the final version.

"The only contribution seems to be the optimization of the acquisition function which is done using stochastic gradient
algorithms" (R2) Our primary contribution is, indeed, an efficient method for optimizing the 2-step optimal acquisition
function. Our secondary contribution is to show that this is practical (i.e., fast enough to use in practice) and provides
more robust performance than both existing widely-used myopic acquisition functions and previously-proposed non-
myopic acquisition functions. While it is true that stochastic gradient ascent is a standard approach, the challenge in
applying it in our setting is in creating an efficient stochatsic gradient estimator and proving it is unbiased.

Other Comments:

"variance on the optimization traces for EI and LCB" (R1): We think this may be because EI and LCB explore less
than other methods, and whether they fall into sub-optimal local optima depends strongly on the problem. Ackley has
more local optima perhaps explaining why performance is less smooth. Define $Q$ (R1,R2): This was a typo. $Q$ is
$2 - \text{OPT}(X_1)$. Code is not provided (R2): We will include our github repo in the final version. Batch evaluations
(R2): Our method generalizes naturally to batch evaluations and our code supports them but we did not include batch
experiments. We'll include them in the appendix in the final version.



[Meta-Review · NeurIPS 2019]

The reviewers mainly agreed that this paper made practically useful and theoretically supported contributions. There was disagreement about the novelty of these contributions. I was convinced by the authors' argument that their central contribution, a stochastic gradient estimator which is both efficient and unbiased, is a worthwhile contribution. In addition, the authors responded convincingly regarding both additional results that they would be including, and comparisons to existing methods. They will also be including source code, and additional experiments on batch experiments in the appendix.